# Age-Stratified Trends in Nutrition and Lifestyle Transitions in Korea: Findings from KNHANES 2013–2022

**DOI:** 10.3390/nu17203282

**Published:** 2025-10-19

**Authors:** Seongryu Bae, Hyuntae Park

**Affiliations:** 1Department of Health Sciences, The Graduate School of Dong-A University, Busan 49315, Republic of Korea; bae.seongryu@gmail.com; 2Digital Healthcare Institute, Dong-A University, Busan 49315, Republic of Korea

**Keywords:** nutrition surveys, dietary intake, meal-related behavior, sedentary behavior, aging

## Abstract

**Background:** Rapid aging and dietary Westernization in Korea have raised concerns about shifting nutrition and lifestyle patterns. **Objectives:** This study examined 10-year trends in nutrient intake, biochemical indicators, meal-related behaviors, and sedentary time among Korean adults, stratified by age group. **Methods**: Data were drawn from the 6th–9th waves (2013–2022) of the Korea National Health and Nutrition Examination Survey (KNHANES) for a cross-sectional analysis, including 61,688 participants (18–39 years: 17,225; 40–64 years: 28,045; ≥65 years: 16,218). Survey-weighted linear and logistic regression analyses were used to test linear trends and between-period contrasts (2013–2015 vs. 2020–2022). **Results**: Total energy intake declined significantly from 2087 kcal in 2013–2015 to 1823 kcal in 2022 (*p* for trend < 0.001), accompanied by a decrease in carbohydrate share (62.8% to 58.3%, *p* < 0.001) and increases in protein (13.8% to 15.4%, *p* < 0.001) and fat (19.0% to 23.4%, *p* < 0.001). Saturated fat and cholesterol intake increased significantly, particularly among older adults (+50.9% and +46.4%, respectively; both *p* < 0.001). The proportion of breakfast skippers increased significantly, more than 35% in 2022 (*p* < 0.001). Conversely, the frequency of eating out decreased (*p* < 0.001), with a particularly pronounced decline during the COVID-19 period. Sedentary time steadily and significantly increased over the decade (*p* < 0.001). **Conclusions**: Over the past decade, Korean adults have shifted toward lower energy and carbohydrate intake, higher protein and fat consumption, and more sedentary lifestyles. Differences in vulnerability by age emphasize the need for personalized public health strategies: promoting protein intake and reducing sedentary time for older adults, while improving dietary patterns and managing weight for younger generations.

## 1. Introduction

Over the past several decades, rapid demographic and epidemiological shifts have reorganized the health care environment in Korea. Korea is one of the most rapidly aging countries in the world, with the elderly population predicted to account for over 40% of the total population by 2050 [1]. These population demographic shifts may contribute to profound changes in lifestyle, dietary habits, and disease burden. The traditional Korean diet, once characterized by high vegetable intake, is gradually shifting toward Westernized eating patterns that include more fat and ultra-processed foods [2,3]. Simultaneously, the prevalence of sedentary lifestyles has increased due to urbanization, digitalization, and the shift toward office-based occupations [4]. These changes have significant implications for chronic diseases such as obesity, diabetes mellitus, and cardiovascular disease which are strongly influenced by both nutritional intake and lifestyle habits [5,6].

Large-scale data such as the Korean National Health and Nutrition Examination Survey (KNHANES) play a critical role in monitoring these health transitions. Since its start in 1998, KNHANES has provided nationally representative data on dietary patterns, health behaviors, and clinical indicators, and has been instrumental in establishing national dietary guidelines, developing chronic disease prevention policies, and designing health promotion programs [7,8]. Studies on dietary and lifestyle-related health indicators using KNHANES data for Koreans are gradually increasing. Several studies have reported long-term dietary changes, including reduced carbohydrate intake and increased fat intake [9,10]. Other studies have investigated health behaviors, such as skipping breakfast, which are becoming increasingly common among young adults and are associated with negative metabolic outcomes [11]. Sedentary behavior, another important lifestyle factor, has also been linked to obesity, chronic diseases, and mortality [12,13].

Previous studies from various countries have provided important insights into recent shifts in dietary and lifestyle patterns. In Japan, analyses from the 2003–2019 National Health and Nutrition Survey demonstrated a clear dietary pattern transition, characterized by a gradual decline in carbohydrate intake and increases in fat and protein consumption, alongside improvements in overall diet quality [14]. Similarly, data from the China Health and Nutrition Survey revealed substantial changes in macronutrient composition between 1991 and 2011, with notable increases in total fat and animal-source foods and a decline in carbohydrate intake, largely driven by urbanization and socioeconomic development [15]. European cohort studies have also documented trends toward higher fat and protein intake and lower carbohydrate intake over time [16].

Despite these global findings, there remains limited evidence on how such nutritional and behavioral transitions have occurred across different generations in Korea. Furthermore, the absence of integrated, age-stratified analyses limits our understanding of how these changes differ across generations and whether recent events such as the COVID-19 pandemic may have accelerated or altered these trends. Reducing this gap is crucial for developing personalized public health strategies and age-specific interventions to promote healthy aging and reduce the risk of chronic diseases [17,18].

Therefore, this study analyzed the 10-year temporal trends in nutrient intake, biochemical indicators, meal-related behaviors, and sedentary time, comparing these across young, middle-aged, and older adults. It also examined both long-term patterns and recent changes by contrasting results from the initial survey period with the most recent survey period.

## 2. Materials and Methods

### 2.1. Study Design and Participants

This observational study was conducted in accordance with the STROBE (Strengthening the Reporting of Observational Studies in Epidemiology) guidelines for cross-sectional studies [19]. The completed STROBE checklist is provided in the Appendix A. This study utilized data from the 6th to 9th waves (2013–2022) of the Korea National Health and Nutrition Examination Survey (KNHANES), a nationwide program implemented by the Ministry of Health and Welfare [8]. KNHANES has been a critical instrument for establishing national public health policies and improving the health status of the Korean population. Beyond its domestic significance, this survey provides essential evidence-based data for international researchers and policymakers. Access to the dataset is available through the official portal (https://knhanes.kdca.go.kr/knhanes/main.do, accessed on 15 June 2025). Survey years were pooled into 3-year cycles to stabilize estimates and align with KNHANES reporting. This study period spanned 10 years, with participant numbers for each time frame as follows: 2013–2015: 18,047; 2016–2018: 19,417; 2019–2021: 18,704; 2022: 5320. Participants with missing data on key variables (dietary intake, biochemical markers, and behavioral indicators) were excluded. Furthermore, for age-specific comparisons, participants were categorized into young adults (18–39 years, n = 17,225), middle-aged adults (40–64 years, n = 28,045), and older adults (≥65 years, n = 16,218).

The data used in this study were approved by the Institutional Review Board of the Korean Centers for Disease Control and Prevention (2013-07CON-03-4C, 2013-12EXP-03-5C, 2018-01-03-P-A, 2018-01-03-C-A, 2018-01-03-2C-A, 2018-01-03-5C-A, 2018-01-03-4C-A), and written informed consent was obtained from all participants.

### 2.2. Participant Characteristics

Household income was divided into quartiles using standardized income data from the KNHANES sample and the overall population (lowest, second, third, and highest quartile). Living arrangement was obtained from the KNHANES household interview. Each respondent reports a complete household roster and household size. We operationalized living alone as residing in a single-person household (yes or no). Blood pressure was measured twice on the right arm and the average value was used. Body mass index (BMI) was calculated as body weight (kg) divided by the square of height (m) [20], with both height and weight measured directly by trained examiners according to standardized KNHANES protocols. Waist circumference (cm) was measured using a tape measure at the midpoint between the lower edge of the last palpable rib and the top of the iliac crest, with the participant standing after a normal exhalation.

### 2.3. Dietary Assessment

Nutrient intake data were collected through face-to-face interviews conducted by trained professionals, including registered dietitians. Dietary information was obtained using a single 24 h recall, in which participants reported all foods and beverages consumed during the preceding day, as well as portion sizes and preparation methods. Estimates of nutrient and energy intakes—such as total energy, carbohydrates, protein, fat, cholesterol, saturated fatty acids, and dietary fiber—were calculated with the Can-Pro 2.0 software (Korean Dietetic Association, Seoul, Republic of Korea) [21]. This validated program, widely applied in both clinical and epidemiological nutrition research in Korea, incorporates a comprehensive database of Korean foods and their nutrient composition. Total energy intake (kcal) and the proportions of macronutrients (carbohydrate, protein, and fat, %) were assessed, along with nutrient intakes expressed per 1000 kcal, including saturated fatty acids (g/1000 kcal) and cholesterol (mg/1000 kcal). To account for individual variations in total energy intake, the nutrient density method was applied, standardizing nutrient values relative to energy intake.

### 2.4. Biochemical Markers

Biochemical measurements were obtained from venous blood after an overnight fast of ≥8 h as per KNHANES procedures [22,23]. The samples were promptly processed, refrigerated, and delivered to the central laboratory of the Seegene Medical Foundation in Seoul. Biochemical parameters, including fasting glucose, hemoglobin A1c, total cholesterol, and triglycerides were analyzed using enzymatic methods with an automated analyzer (Hitachi 7600 Automatic Analyzer, Hitachi, Tokyo, Japan).

### 2.5. Behavioral Lifestyle Factors

Meal-related behaviors were evaluated in terms of breakfast consumption and frequency of eating out. Breakfast consumption was assessed by asking participants about their average weekly frequency of breakfast over the past year. Participants who reported eating breakfast three or more times per week were classified as regular consumers, whereas those consuming breakfast two or fewer times per week were classified as breakfast skippers [24]. Frequency of eating out was determined based on participants’ self-reported average number of meals consumed outside the home during the previous year. Participants were categorized as frequent eaters-out if they ate out two or more times per week, and as rare eaters-out if they reported eating out less than once per month [25].

Sedentary behavior was evaluated using a self-administered questionnaire. Participants were asked: “On average, how many hours per day do you spend sitting?” This question encompassed sedentary time across multiple contexts, including work, home, academic activities, and leisure. Examples provided were sitting at a desk, socializing, reading, or sitting/lying down to watch television. Participants reported their estimated daily sedentary time, which was recorded in hours per day.

### 2.6. Statistical Analysis

All estimates accounted for the complex sample design. Component-specific sampling weights were applied: the nutrition weight for 24 h recall variables and diet behaviors, the examination weight for anthropometry and biochemical markers, and the household/interview weight for socio-demographic characteristics. For each period (no age stratification), survey-weighted means and 95% confidence intervals (CIs) were reported for continuous variables. For binary outcomes, survey-weighted prevalences (%) were presented with logit-based 95% CIs. For continuous outcomes, we fit survey-weighted linear regressions with the outcome as the dependent variable and period as the single ordinal predictor; the Wald test for the period coefficient provided the *p* value for linear trend. For binary outcomes, we used survey-weighted logistic regressions with period entered as an ordinal predictor and obtained the *p* for trend from the corresponding Wald test. Between-age differences were evaluated within each period using the same survey-weighted modeling framework, specifying age group as a categorical predictor and estimating pairwise marginal contrasts (young vs. middle, young vs. older, middle vs. older). For continuous outcomes, differences in weighted means with 95% CIs were reported; for binary outcomes, differences in weighted prevalence with 95% CIs were reported. To control family-wise type-I error for the three pairwise age contrasts within a period and variable, Bonferroni adjustment was applied to Wald test *p*-values. To summarize change between 2013–2015 and 2020–2022, survey-weighted regression provided period contrasts. For continuous outcomes, results are presented as absolute mean differences and percent change ((late/early) − 1) × 100. For binary outcomes, percent change in weighted prevalence was computed analogously. CIs and *p*-values for these contrasts derive from the model-based Wald inference. Analyses were conducted using R (version 4.5.1; Foundation for Statistical Computing, Vienna, Austria) and SPSS (version 28.0; IBM Corp., Armonk, NY, USA). Statistical significance was set a priori at *p* < 0.05.

## 3. Results

### 3.1. Population Characteristics and Overall Trend

Table 1 presents the results showing temporal trends across survey cycles from 2013 to 2022. From 2013 to 2022, the distribution of sex remained relatively stable, with men accounting for approximately 44% and women 56% of the total population, without significant temporal changes. The age distribution showed a gradual trend toward aging, and the weighted proportion of people living alone showed an increasing trend. Systolic blood pressure (SBP), BMI, and waist circumference also showed a trend of gradual increase (trend *p*-values all <0.001). Total daily energy intake decreased significantly from 2087 kcal in 2013–2015 to 1823 kcal in 2022 (*p* < 0.001). Carbohydrate intake decreased from 62.8% to 58.3% of total energy (*p* < 0.001), while protein increased from 13.8% to 15.4% and fat increased from 19.0% to 23.4% (both *p* < 0.001). Saturated fat intake increased from 6.1 g to 8.1 g per 1000 kcal, and cholesterol intake also increased from 118.7 mg to 144.0 mg per 1000 kcal (both *p* < 0.001).

Biochemical markers showed slight changes over the cycle. Fasting blood glucose increased from 99.0 mg/dL to 100.5 mg/dL, HbA1c increased but remained within the 5.6–5.8% range, total cholesterol slightly increased from 187.7 mg/dL to 190.8 mg/dL, and triglycerides decreased from 137.9 mg/dL to 130.4 mg/dL (all *p* < 0.001). In terms of dietary behaviors, breakfast skipping became increasingly prevalent, rising from 25.0% to 35.1% (*p* <0.001), whereas the proportion of individuals eating out declined from 43.8% to 38.6% during the same period (*p* <0.001). Sedentary time, however, has increased significantly, with the average daily sitting time increasing from 7.7 h in 2013–2015 to 8.9 h in 2022.

### 3.2. Age-Related Trajectories of Health Indicators (2013–2022)

Figure 1 illustrates the temporal trends in vital signs and anthropometric measures (Figure 1A), nutrient intake (Figure 1B), biochemical markers (Figure 1C), and lifestyle behaviors (Figure 1D) between 2013 and 2022 across young, middle-aged, and older adults.

Overall, SBP showed a modest upward trend across all age groups, with consistently higher values in older adults compared to younger and middle-aged groups (Figure 1A). In contrast, BMI and waist circumference increased gradually in all age groups, but the gap between younger and older adults narrowed over time, reflecting a relatively steeper rise among younger individuals (Figure 1A).

Nutrient intake demonstrated marked shifts over time. Total energy intake declined steadily in all age groups, but the reduction was most pronounced among young adults, suggesting age-related differences in energy consumption patterns, though cross-sectional data cannot distinguish between age effects and cohort effects (Figure 1B). Carbohydrate intake steadily decreased across all age groups, with the highest levels consistently observed among older adults and the lowest among young adults (Figure 1B). Protein intake gradually increased over time across all age groups, with the highest levels in young adults and the lowest in older adults (Figure 1B). Intake of lipid-related nutrients, including total fat, saturated fatty acids, and cholesterol, showed a consistent upward trend across all age groups (Figure 1B).

Biochemical markers revealed distinct age-related differences (Figure 1C). Fasting glucose and hemoglobin A1c levels were consistently highest in older adults across all survey cycles, indicating an age-related disadvantage in glycemic control. Younger and middle-aged adults maintained comparatively lower values, with relatively stable trajectories over time. Total cholesterol levels showed contrasting trends across age groups. Young adults started at the lowest baseline and exhibited the steepest increase, while middle-aged adults showed a more gradual but steady upward trend. Conversely, older adults initially had the highest levels but tended to decrease over time. Middle-aged adults had the highest triglyceride levels, but these gradually decreased over time. The older adult levels also showed a decrease, while young adults maintained relatively stable levels at the lowest point.

Dietary patterns also showed pronounced differences across age groups. The proportion of young adults skipping breakfast dramatically increased, over 40% by 2022, while middle-aged and older adults showed a relatively steady rise (Figure 1D). Conversely, eating out remained consistently highest among young adults but decreased across all age groups over time, with a particularly marked decline during the 2019–2021 period (Figure 1D). Sedentary time increased across the entire population, but the steepest rise was observed among young adults (Figure 1D).

### 3.3. Age-Stratified Percent Change in Health Indicators Between 2013–2015 and 2020–2022

Figure 2 presents the percent changes in nutritional intake, biochemical markers, anthropometric indices, and lifestyle behaviors between 2013–2015 and 2020–2022 across young, middle-aged, and older adults.

Between 2013–2015 and 2020–2022, total energy intake (Young −11.7%, Middle −9.6%, Old −4.9%) and carbohydrate intake (all groups −7%) decreased significantly across all age groups. In contrast, protein intake increased by 12.2% among young adults, 10.0% among middle-aged adults, and 12.7% among older adults (all *p* < 0.001). Although intake of lipid-related nutrients such as fat, saturated fat, and cholesterol was highest among younger adults, as shown in Figure 1B, the rate of increase was highest among older adults. Between 2013–2015 and 2020–2022, the increase in fat, saturated fat, and cholesterol intake among the older adults was substantial, at 40.9%, 50.9%, and 46.4%, respectively (all *p* < 0.001).

BMI and waist circumference showed a slight increase throughout the lifespan, with a relatively larger proportion of increase observed in young and middle-aged adults. SBP showed a significant increase only in the younger age group (1.7%, *p* < 0.001).

Fasting blood glucose and HbA1c levels remained generally stable across all age groups between 2013–2015 and 2020–2022. Total cholesterol increased in the young (1.5%, *p* < 0.001) and middle-aged (*p* < 0.001) groups, while it decreased significantly (−4.9%, *p* < 0.001) in the older group. Triglycerides showed no significant difference in the young group but decreased significantly in the middle-aged (−5.5%, *p* < 0.001) and older adults (10.1%, *p* < 0.001) groups.

In terms of dietary patterns, the proportion of young adults skipping breakfast showed a significant increase to 16.5% (*p* < 0.001), followed by a high increase among middle-aged adults. Although the increase was smaller among the older adults, it still showed an upward trend (2.1%, *p* < 0.001). The frequency of eating out was originally highest among young adults, but it decreased sharply in 2020–2022 (−6.4%, *p* < 0.001). Conversely, the change was relatively minor among the older adults, who had a low frequency of eating out to begin with (−0.9%, *p* = 0.99). Sedentary time increased significantly across all age groups, with the largest increase observed among the older age group (19.9%, *p* < 0.001). Following this, sedentary time increased among middle-aged adults (14.9%, *p* < 0.001) and young adults (11.6%, *p* < 0.001).

## 4. Discussion

### 4.1. Summary of Key Findings

In this nationally representative study of Korean adults using KNHANES 2013–2022 data, we found that total energy and carbohydrate intake declined steadily across all age groups, while protein, fat, saturated fat, and cholesterol intake increased significantly. Although absolute intake of lipid-related nutrients was highest among younger adults, the relative rate of increase was greatest in older adults. Carbohydrate intake remained highest in older adults and lowest in younger adults, whereas protein intake showed the opposite pattern.

These dietary changes were accompanied by modest increases in BMI, waist circumference, and SBP, with younger adults showing steeper rises than older adults. Biochemical markers displayed age-specific divergences: total cholesterol increased most in younger adults but decreased in older adults, while triglycerides declined in middle-aged and older adults.

Behavioral indicators also shifted markedly. Breakfast skipping rose sharply, particularly among young adults, exceeding 40% by 2022. Eating out, originally most frequent among young adults, showed a pronounced decline, with the steepest reductions during the COVID-19 pandemic. Sedentary time increased across all groups, with the greatest rise in older adults.

### 4.2. Comparison with Previous Studies

This macronutrient rebalancing is directionally consistent with prior Korean trend reports showing a long-run reduction in carbohydrate share and increases in fat and protein as diets Westernize. Analyses of KNHANES 1998–2018 documented declines in grain staples and increases in animal-source foods and fats, with a concomitant shift in macronutrient composition [10]. A more recent secular-trend study (2010–2020) also reported decreasing carbohydrate and rising protein/fat shares [9]. From 1999 to 2016, dietary energy intake among U.S. adults also showed a broadly similar trend. The proportion of low-quality carbohydrates decreased significantly, while the proportion of high-quality carbohydrates, plant-based protein, and polyunsaturated fats increased significantly. Despite improvements in key nutrient composition and diet quality, persistently high intakes of low-quality carbohydrates and saturated fat remained [26]. Moreover, longitudinal cohort studies in Europe have also confirmed shifts toward reduced carbohydrate intake and increased fat and protein consumption. For example, Winkvist et al. (a 10-year Swedish cohort) reported that dietary changes over a decade were characterized by decreased carbohydrate intake and increased protein and fat intake [16]. Similarly, the Amsterdam Growth & Health Longitudinal Study documented significant temporal effects on energy intake and the composition of major nutrients (protein, fat, carbohydrates) during a 20-year follow-up period [27]. Several factors may help explain the age-specific patterns we observed. One possibility is the greater penetration of ultra-processed foods (UPFs) and animal-source convenience foods among younger cohorts, which is consistent with evidence linking higher UPFs consumption to greater obesity risk, particularly among Korean women [28]. The NOVA framework further clarifies how food processing alters dietary fat and sugar profiles, thereby increasing energy density. Monteiro et al. (2019) [29] classified foods into four categories—unprocessed or minimally processed foods, processed culinary ingredients, processed foods, and UPFs—and highlighted that UPFs are typically high in added fats, refined sugars, sodium, and food additives, while being low in dietary fiber, protein quality, and micronutrients. The authors emphasized that the widespread consumption of UPFs shifts overall dietary patterns toward higher intakes of saturated fat and free sugars and lower nutritional quality, which in turn contributes to excessive energy intake, obesity, and metabolic disorders [29]. In addition, the increasing trend in fat, saturated fat, and cholesterol intake among the older adults may reflect generational shifts in dietary habits. This suggests that Korean seniors, who traditionally relied more on carbohydrate-centered diets, are gradually adopting Westernized diets rich in animal fats, dairy products, and high-fat processed foods. This pattern aligns with findings from Song et al. (2019), who reported a steady increase in total fat and saturated fatty acid intake among Korean adults over time [30]. Similarly, nationwide trend data from Oh et al. (2025) indicate that dietary quality among Korean seniors deteriorated between 2013 and 2022, characterized by increased intake of lipid density and cholesterol [31]. These findings suggest that the older adults, who already have greater baseline cardiovascular metabolic risk, may be experiencing compounding vulnerability as dietary lipid load increases during a period of declining metabolic resilience.

Our findings demonstrate clear generational contrasts in macronutrient consumption: carbohydrate intake was consistently highest in older adults and lowest in younger adults, whereas protein intake followed the opposite pattern. However, these age-related differences may reflect both generational dietary preferences and age-related physiological changes. This pattern carries important health implications. In older adults, insufficient protein intake is concerning given the well-documented age-related decline in skeletal muscle mass and strength, known as sarcopenia. Given evidence from the PROT-AGE Study Group, which recommends daily protein intakes of 1.0–1.2 g/kg body weight in older adults to support muscle maintenance and function, lower relative protein intake in older Koreans is especially concerning [32,33]. Furthermore, a meta-analysis by Coelho-Júnior et al. (2022) showed higher protein consumption above the Recommended Daily Allowance correlates with better physical performance and strength in older populations [34]. These findings suggest that older adults, already at greater baseline cardiometabolic risk, may now face compounded vulnerability due to both deteriorating diet quality (higher saturated fat/cholesterol) and insufficient protein relative to emerging recommendations. Young adults in our cohort exhibited the highest protein and lowest carbohydrate intake, which appears to reflect weight management motivation, fitness-focused lifestyles, and widespread exposure to Westernized dietary norms. Moderate carbohydrate reduction while maintaining adequate protein quality may help reduce weight and body fat while contributing to muscle mass preservation, as demonstrated in meta-analyses and reviews of high-protein diets [35]. However, large prospective cohort studies and meta-analyses have identified an association between very low carbohydrate intake and increased long-term mortality [36]. This data suggests that while young adults may gain metabolic benefits from relatively high protein intake, a balanced macronutrient source remains crucial for long-term health.

BMI and waist circumference showed a gradual overall increase, with a relatively larger proportion of increase observed among younger and middle-aged adults. SBP increased slightly but significantly only in the younger age group. These findings are consistent with reports that the average SBP among Korean adults remained stable over the past decade despite changes in obesity patterns [37]. Although absolute values for SBP and waist circumference remain high in the older age group, this suggests that weight management and blood pressure control should be prioritized for prevention in early adulthood.

While fasting blood glucose and HbA1c remained mostly stable, total cholesterol increased most significantly in young adults but decreased in the older adults; triglycerides decreased in middle-aged and older adults but did not decrease in younger adults. These patterns show consistency biologically and policy-wise. In Korea, statin treatment coverage and LDL-C target attainment rates increased substantially throughout the 2010s, with statins accounting for over 90% of lipid-lowering drug prescriptions by 2018. This prevalence is highest among the older adults [38]. A related analysis of US NHANES data also showed declines in average total cholesterol, LDL-C, and triglyceride levels at the population level from 1999 to 2018, consistent with the expansion of statin use [39]. Therefore, the phenomenon of decreased total cholesterol levels in the older adults despite increased saturated fat and cholesterol intake can be explained by the counterbalancing effect of treatment and secondary prevention intensity on dietary changes. Conversely, younger adults with lower statin use may have shown a tendency toward increased total cholesterol levels alongside rising fat/saturated fat intake.

In this study, meal-related behaviors showed a significant increase in skipping breakfast, particularly among young adults. This finding aligns with research indicating that reduced breakfast frequency in young adults is associated with an increased risk of developing metabolic syndrome and that skipping breakfast and irregular eating patterns elevate metabolic risk [40]. In the context of this study, the increasing phenomenon of skipping breakfast among young adults may occur alongside rising BMI and increased high-fat dietary intake. This underscores the need to improve breakfast patterns, along with overall dietary quality, in this younger age group. Furthermore, the frequency of eating out decreased across all age groups, with a particularly pronounced decline among young adults between 2019 and 2021. Further studies confirmed that during the COVID-19 period, there was reduced restaurant patronage and a shift toward at-home eating among young adults, along with decreased frequency of eating out [41]. The period 2020–2022, corresponding to the COVID-19 pandemic, was analyzed separately to account for its potential confounding effects. Behavioral indicators such as eating-out frequency and sedentary time showed more pronounced changes during this period, reflecting pandemic-related restrictions, whereas macronutrient composition trends appeared to follow longer-term secular shifts.

This study demonstrated a consistent increase in sedentary time over a decade. Young adults consistently reported the highest levels, while older adults showed a marked increase in recent years. These findings align with a KNHANES-based analysis showing an increase in obesity prevalence among adults sitting for 8 h or more per day between 2014 and 2021 [4]. Long sitting time was associated with increased all-cause mortality and cardiovascular mortality risk even after adjusting for physical activity [12]. In addition, the Association of 12-Year Trajectories of Sitting Time with Frailty study found that among older adults, greater cumulative sedentary time over a decade was significantly associated with increased frailty [42]. In younger populations, the European Youth Heart Study demonstrated that higher screen time in adolescence predicted elevated adiposity, triglycerides, and metabolic syndrome scores in young adulthood [43].

### 4.3. Strengths and Limitation

The strengths of this study include the use of large-scale, nationally representative data spanning a 10-year period, which enabled robust estimation of trends in nutrient intake, biochemical markers, meal-related behaviors, and sedentary time. The stratified analysis by age groups offers novel insights into generational disparities that have not been well documented in previous Korean studies. Furthermore, the integration of both dietary and behavioral indicators provides a holistic picture of population health trajectories. A key methodological strength of this study is that dietary intake data were collected through face-to-face interviews conducted by trained professionals, including registered dietitians, which ensures high data quality and measurement reliability and minimizes common sources of error such as misinterpretation of questions or inaccurate portion size estimation.

Nevertheless, several limitations of this study should be acknowledged. First, the use of cross-sectional KNHANES data limits the ability to establish causal relationships between dietary or behavioral changes and health outcomes. Second, the observed age-group differences may partly reflect cohort effects, as distinct historical and socioeconomic conditions experienced by different birth cohorts, rather than true generational changes, could have influenced the findings. Third, survival bias is possible among older adults, since healthier individuals are more likely to participate in surveys. Fourth, dietary intake was assessed using a single 24 h recall, which may not fully capture habitual intake and is subject to recall bias and intra-individual variation. In addition, sedentary behavior was assessed using a single self-reported question rather than a validated multi-item instrument such as the International Physical Activity Questionnaire (IPAQ), which may have introduced some degree of measurement bias and may have resulted in underestimation or overestimation of actual sedentary time. Future studies would benefit from incorporating comprehensive validated instruments to capture more nuanced patterns of sedentary behavior across different contexts and time periods. Finally, potential confounders such as medication use (e.g., statins, antihypertensives), socioeconomic transitions, and other cohort-related factors were not fully accounted for, which may have affected biochemical marker trends. Future longitudinal designs or prospective cohort studies will be essential to clarify temporal and causal associations.

## 5. Conclusions

Our findings indicate that the dietary and lifestyle habits of Koreans are rapidly shifting toward Westernized patterns and prolonged sedentary lifestyles, with distinct age-related vulnerabilities. The older population is disproportionately affected by increasing lipid intake and sedentary time, while young adults are increasingly exposed to lifestyle risks through skipping breakfast, reduced energy intake, and weight gain. These findings emphasize the importance of developing generation-specific public health strategies that suggest the need for age-specific public health approaches that consider observed dietary and behavioral patterns. However, longitudinal studies are needed to distinguish true generational effects from age-related changes before implementing targeted interventions for chronic disease prevention in this aging population.

## Figures and Tables

**Figure 1 nutrients-17-03282-f001:**
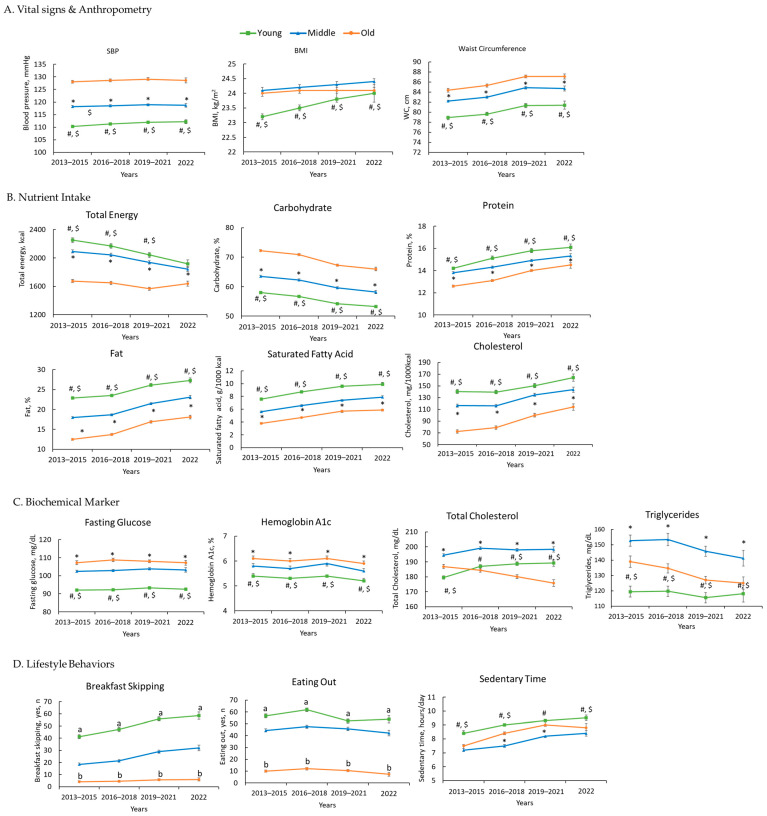
Ten-Year Trends in Health Indicators Across Young, Middle-Aged, and Older Adults: KNHANES 2013–2022. (**A**) Vital Signs and Anthropometry, (**B**) Nutrient Intake, (**C**) Biochemical Marker, (**D**) Lifestyle Behaviors. The values for each variable are presented as weighted means (95% CI) and weighted rate (95% CI). Superscript letters denote post hoc comparisons: #, young vs. middle-aged; $, young vs. old; *, middle-aged vs. old. a, statistically significant association by adjusted standardized residual > 1.96 (*p* < 0.05). b, statistically significant association by adjusted standardized residual < −1.96 (*p* < 0.05). SBP, systolic blood pressure; BMI, body mass index.

**Figure 2 nutrients-17-03282-f002:**
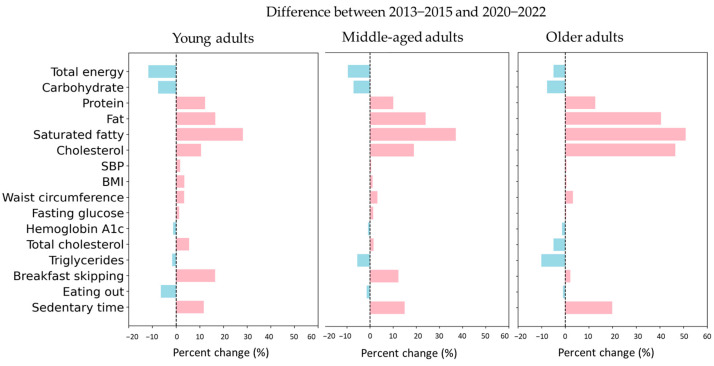
Percent changes in health indicators between 2013–2015 and 2020–2022 across young, middle-aged, and older adults. Bars represent percent change calculated from weighted means or weighted rate using complex survey weights. SBP, systolic blood pressure; BMI, body mass index.

**Table 1 nutrients-17-03282-t001:** Population characteristics and overall trend of KNHANES between 2013 and 2022.

Variables	Total	2013–2015	2016–2018	2019–2021	2022	*p* for Trend
Overall, n	61,488	18,047	19,417	18,704	5320	
Sex, weighted % (95% CI)						0.286
Male	44.2 (43.8 to 44.7)	43.7 (42.9 to 44.4)	44.3 (43.6 to 45.1)	45.0 (44.2 to 45.7)	43.4 (42.0 to 44.8)	
Female	55.8 (55.3 to 56.2)	56.3 (55.6 to 57.1)	55.7 (54.9 to 56.4)	55.0 (54.3 to 55.8)	56.6 (55.2 to 58.0)	
Age group, weighted % (95% CI)						<0.001
Young adults (18–39 y)	28.7 (28.3 to 29.1)	30.4 (29.7 to 31.1)	29.6 (28.9 to 30.3)	27.0 (26.3 to 27.7)	26.3 (25.1 to 27.6)	
Middle-aged (40–64 y)	45.7 (45.3 to 46.1)	45.4 (44.6 to 46.2)	45.7 (44.9 to 46.4)	46.1 (45.4 to 46.9)	45.0 (44.2 to 45.7)	
Older adults (≥65 y)	25.6 (25.2 to 26.0)	24.2 (23.5 to 24.8)	24.7 (24.0 to 25.3)	26.9 (26.2 to 27.6)	28.6 (27.3 to 29.9)	
Household income, weighted % (95% CI)						<0.001
Lowest quartile	29.0 (28.6 to 29.4)	27.5 (26.8 to 28.2)	28.8 (28.1 to 29.5)	30.2 (29.5 to 30.9)	30.0 (28.7 to 31.3)	
Second quartile	27.4 (27.0 to 27.8)	27.2 (26.6 to 27.9)	27.5 (26.8 to 28.2)	27.3 (26.6 to 28.0)	27.7 (26.4 to 29.0)	
Third quartile	24.7 (24.3 to 25.0)	25.5 (24.8 to 26.2)	24.6 (23.9 to 25.2)	24.3 (23.6 to 25.0)	23.4 (22.2 to 24.6)	
Highest quartile	19.0 (18.7 to 19.3)	19.7 (19.1 to 20.3)	19.1 (18.5 to 19.7)	18.2 (17.7 to 18.7)	18.9 (17.8 to 20.0)	
Living along, weighted % (95% CI)						<0.001
Yes	12.1 (11.9 to 12.4)	9.5 (9.1 to 10.0)	12.1 (11.6 to 12.6)	13.5 (13.0 to 14.1)	15.6 (14.6 to 16.6)	
Vital signs & Anthropometry, weighted mean (95% CI)						
SBP, mmHg	117.7 (117.5 to 117.8)	116.7 (116.4 to 117.0)	117.6 (117.3 to 117.8)	118.4 (118.2 to 118.7)	118.6 (118.1 to 119.1)	<0.001
BMI, kg/m^2^	24.0 (23.9 to 24.0)	23.7 (23.7 to 23.8)	23.9 (23.9 to 24.0)	24.1 (24.0 to 24.2)	24.2 (24.1 to 24.3)	<0.001
Waist circumference, cm	82.7 (82.6 to 82.8)	81.3 (81.1 to 81.4)	82.1 (82.0 to 82.3)	84.1 (83.9 to 84.3)	84.1 (83.8 to 84.5)	<0.001
Nutrient intake, weighted mean (95% CI)						
Total energy, kcal	1984 (1975 to 1993)	2087 (2070 to 2105)	2023 (2007 to 2040)	1904 (1888 to 1920)	1823 (1798 to 1848)	<0.001
Carbohydrate, %	60.8 (60.7 to 61.0)	62.8 (62.5 to 63.0)	61.6 (61.4 to 61.9)	59.2 (58.9 to 59.4)	58.3 (57.8 to 58.7)	<0.001
Protein, %	14.5 (14.5 to 14.6)	13.8 (13.7 to 13.8)	14.4 (14.3 to 14.5)	15.1 (15.0 to 15.1)	15.4 (15.3 to 15.5)	<0.001
Fat, %	20.6 (20.5 to 20.7)	19.0 (18.8 to 19.2)	19.6 (19.4 to 19.8)	22.2 (22.1 to 22.4)	23.4 (23.1 to 23.7)	<0.001
Saturated fatty, g/1000 kcal	7.1 (7.1 to 7.2)	6.1 (6.0 to 6.2)	7.1 (7.0 to 7.2)	7.9 (7.8 to 7.9)	8.1 (8.0 to 8.3)	<0.001
Cholesterol, mg/1000 kcal	125.9 (124.9 to 126.9)	118.7 (116.8 to 120.6)	118.5 (116.9 to 120.2)	133.7 (131.8 to 135.6)	144.0 (140.8 to 147.1)	<0.001
Biochemical marker, weighted mean (95% CI)						
Fasting glucose, mg/dL	100.1 (99.8 to 100.3)	99.0 (98.6 to 99.4)	99.9 (99.5 to 100.3)	100.9 (100.6 to 101.3)	100.5 (99.8 to 101.2)	<0.001
Hemoglobin A1c, %	5.7 (5.7 to 5.7)	5.7 (5.7 to 5.7)	5.6 (5.6 to 5.6)	5.8 (5.7 to 5.8)	5.6 (5.5 to 5.6)	<0.001
Total cholesterol, mg/dL	190.7 (190.3 to 191.0)	187.7 (187.0 to 188.3)	192.4 (191.8 to 193.0)	191.6 (190.9 to 192.3)	190.8 (189.6 to 192.0)	<0.001
Triglycerides, mg/dL	135.3 (134.1 to136.5)	137.9 (135.6 to 140.2)	138.1 (135.8 to 140.4)	131.9 (129.9 to 133.9)	130.4 (127.3 to 133.5)	<0.001
Meal-related behavior, weighted % (95% CI)						
Breakfast skipping, yes	29.6 (29.2 to 30.1)	25.0 (24.2 to 25.8)	28.0 (27.1 to 28.8)	33.9 (33.0 to 34.8)	35.1 (33.6 to 36.7)	<0.001
Frequent eating out, yes	43.4 (42.9 to 43.9)	43.8 (42.9 to 44.7)	46.9 (46.0 to 47.8)	41.5 (40.5 to 42.4)	38.6 (37.0 to 40.2)	<0.001
Physical activity, weighted mean (95% CI)						
Sedentary time, hours/day	8.3 (8.3 to 8.4)	7.7 (7.6 to 7.8)	8.2 (8.2 to 8.3)	8.7 (8.6 to 8.8)	8.9 (8.7 to 9.0)	<0.001

Values are presented as weighted mean (95% CI) or weighted rate (95% CI). CI, confidence interval; SBP, systolic blood pressure; BMI, body mass index.

## Data Availability

This study analyzed data released from government agencies: [https://knhanes.kdca.go.kr] (accessed on 15 June 2025).

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
