# Peer review of "Age-Stratified Trends in Nutrition and Lifestyle Transitions in Korea: Findings from KNHANES 2013–2022"

_nutrients, 2025, doi:10.3390/nu17203282_

Round 1

Reviewer 1 Report

Comments and Suggestions for Authors

Comments to authors

This study analyses changes in the diets, eating behaviours and sedentary lifestyles of Korean adults between 2013 and 2022, categorised by age group. The findings reveal a shift towards lower carbohydrate intake and higher protein and fat consumption, alongside increased sedentary behaviour and breakfast skipping. While the authors have made an effort to produce a comprehensive and coherent article, several improvements are required before it can be considered for publication.

Abstract

  • It would be helpful to include a brief introduction.
  • In the Methods section, a line should be added detailing the statistical analyses performed. ANOVAs? Or what analysis was performed?
  • In the provided data (pre- and post-data), it would also be helpful to include the p-value to determine whether there are statistically significant differences between the before and after results.

Introduction

  • “These changes have significant implica-42 tions for chronic diseases such as obesity, diabetes, and cardiovascular disease which are 43 strongly influenced by both nutritional intake and lifestyle habits [5,6]. 44”: change diabetes by diabetes mellitus.
  • “of how these changes differ across generations and 61 whether recent events like the pandemic may have accelerated or altered these trends. 62”: Covid pandemic, I imagine.
  • Lines 57–65: The authors have not cited this paragraph. I interpret it as a combination of hypothesis and consequences. I suggest adding a citation at the end of the paragraph, even if it's just a reference to lines 63–65.

Methods

  • An important point: as this is an original observational study, there are likely to be guidelines for conducting this type of study. It may be STROBE, but authors should always look for the guidelines that best fit their study. In any case, this should be noted at the beginning of the 'Methods' section, and the completed STROBE (or other applicable) checklist should be included in the supplementary material.
  • “Body mass index 96 (BMI) was calculated as body weight (kg) divided by the square of height (m).”: Although the reader knows how BMI is calculated, a citation should be included.
  • “2.4. Biochemical Markers”: Were all of these parameters determined on fasting? If so, please specify.

Results

  • Lines 166-187: You should split the paragraph into two. Paragraphs this long are difficult to read.
  • Table 1. Table titles usually go above the table, not below.
  • “This suggests that despite an 205 overall decline in carbohydrate consumption, older adults still depend more heavily on 206 carbohydrates compared to younger generations.”, “These trends highlight generational differences in the 210 consumption of macronutrients, suggesting that young adults are shifting toward high-211 protein, low-carbohydrate diets more rapidly than the older adults.”, etc etc. Results should describe results, not interpretations or hypotheses; that's what the discussion is for. This is important for two reasons. First, as I said, results should focus on the obtained data. Second, eliminating information that belongs in other sections makes this section easier to read. Review the results section to ensure that all text belongs there and not in the discussion section.
  • Likewise, the authors begin with “Figure 1 illustrates the temporal trends in vital signs,”. However, it would be appreciated if, throughout the text, you could include “Figure 1-A”, “Figure 1-B”, etc., to help the reader find their bearings more easily.

Discussion

The authors did an excellent job in the discussion; it's clear they've tried to explain all the results obtained. I only have two questions.:

  • “4.1. Summary of Key Findings”: This section is too long. It should summarise the main findings of your study in no more than 15 lines.
  • It would be interesting to include a paragraph (no more than 12–15 lines) explicitly comparing your results with those of other world regions (America, Europe, etc.). In other words, it would be interesting to see whether the same results are obtained in other countries.

Author Response

We sincerely appreciate the time and effort you have dedicated to reviewing our work and for providing valuable insights to enhance the quality and impact of our study. Responses to the reviewers' comments and changes in the revised manuscript have been highlighted in red.

Reviewer 1 Comments

This study analyses changes in the diets, eating behaviours and sedentary lifestyles of Korean adults between 2013 and 2022, categorised by age group. The findings reveal a shift towards lower carbohydrate intake and higher protein and fat consumption, alongside increased sedentary behaviour and breakfast skipping. While the authors have made an effort to produce a comprehensive and coherent article, several improvements are required before it can be considered for publication.

Response

We sincerely appreciate the time and effort you have dedicated to reviewing our work and for providing valuable insights to enhance the quality and impact of our study. Responses to the reviewers' comments and changes in the revised manuscript have been highlighted in red.

Abstract

Comment 1

It would be helpful to include a brief introduction.

Response 1

We appreciate this suggestion. In the revised Abstract, we have incorporated a concise background sentence highlighting Korea’s rapid demographic shifts and Westernization of dietary habits, to better contextualize the study objectives.

Line 10-11

“Background: Rapid aging and dietary Westernization in Korea have raised concerns about shifting nutrition and lifestyle patterns.”

Comment 2

In the Methods section, a line should be added detailing the statistical analyses performed. ANOVAs? Or what analysis was performed?

Response 2

Thank you for this important comment. We have revised the Methods portion of the Abstract to explicitly specify that survey-weighted linear and logistic regression analyses were performed to evaluate trends and between-period contrasts.

Line 17-18

“Survey-weighted linear and logistic regression analyses were used to test linear trends and between-period contrasts (2013–2015 vs. 2020–2022).”

Comment 3

In the provided data (pre- and post-data), it would also be helpful to include the p-value to determine whether there are statistically significant differences between the before and after results.

Response 3

We completely agree that including p-values in the Abstract strengthens the presentation of our findings. In the revised Abstract, we have added p-values for the main findings to clarify the statistical significance of temporal changes.

Line 18-26

“Total energy intake declined significantly from 2087 kcal in 2013–2015 to 1823 kcal in 2022 (p for trend < 0.001), accompanied by a decrease in carbohydrate share (62.8% to 58.3%, p < 0.001) and increases in protein (13.8% to 15.4%, p < 0.001) and fat (19.0% to 23.4%, p < 0.001). Saturated fat and cholesterol intake increased significantly, particularly among older adults (+50.9% and +46.4%, respectively; both p < 0.001). The proportion of breakfast skippers increased significantly, more than 35% in 2022 (p < 0.001). Conversely, the frequency of eating out decreased (p < 0.001), with a particularly pronounced decline during the COVID-19 period. Sedentary time steadily and significantly increased over the decade (p < 0.001).”

Comment 4

“These changes have significant implications for chronic diseases such as obesity, diabetes, and cardiovascular disease which are strongly influenced by both nutritional intake and lifestyle habits [5,6].”: change diabetes by diabetes mellitus.

Response 4

We appreciate the reviewer’s careful suggestion. In the revised manuscript, we have replaced “diabetes” with the more precise term “diabetes mellitus” to clinical accuracy and consistency with standard medical nomenclature in the Introduction section (line 45).

Comment 5

“of how these changes differ across generations and whether recent events like the pandemic may have accelerated or altered these trends.”: Covid pandemic, I imagine.

Response 5

Thank you for pointing this out. We agree that specificity improves clarity, and have revised the wording to use “COVID-19 pandemic” instead of the less precise term “pandemic” throughout the manuscript.

Comment 6

Lines 57–65: The authors have not cited this paragraph. I interpret it as a combination of hypothesis and consequences. I suggest adding a citation at the end of the paragraph, even if it's just a reference to lines 63–65.

Response 6
We appreciate this insightful comment. To strengthen the paragraph and provide appropriate scholarly support for our statements, we have added relevant reference at the end of the paragraph (line 76), citing recent literature that discusses age-stratified dietary and lifestyle transitions in Korea.

Added Reference

  1. Jo, G.; Park, D.; Lee, J.; Kim, R.; Subramanian, S.V.; Oh, H.; Shin, M.J. Trends in Diet Quality and Cardiometabolic Risk Factors Among Korean Adults, 2007-2018. JAMA Netw Open 2022, 5, e2218297, doi:10.1001/jamanetworkopen.2022.18297.
  2. Yang, H.J.; Park, S.; Yoon, T.Y.; Ryoo, J.H.; Park, S.K.; Jung, J.Y.; Lee, J.H.; Oh, C.M. Nationwide changes in physical activity, nutrient intake, and obesity in South Korea during the COVID-19 pandemic era. Front Endocrinol (Lausanne) 2022, 13, 965842, doi:10.3389/fendo.2022.965842.

Methods

Comment 7

An important point: as this is an original observational study, there are likely to be guidelines for conducting this type of study. It may be STROBE, but authors should always look for the guidelines that best fit their study. In any case, this should be noted at the beginning of the 'Methods' section, and the completed STROBE (or other applicable) checklist should be included in the supplementary material.

Response 7

We thank the reviewer for this valuable suggestion. We agree that adherence to established reporting guidelines is important for observational studies and enhances the transparency and reproducibility of our research. Accordingly, we have revised the beginning of the Methods section to state that the study was conducted in accordance with the STROBE (Strengthening the Reporting of Observational Studies in Epidemiology) guidelines for cross-sectional studies. Furthermore, the completed STROBE checklist has been prepared and will be uploaded as Supplementary Material, as recommended to demonstrate compliance with international reporting standards.

Line 84-87
“This observational study was conducted in accordance with the STROBE (Strengthening the Reporting of Observational Studies in Epidemiology) guidelines for cross-sectional studies. The completed STROBE checklist is provided in the Supplementary Material.”

Supplementary Table S1. STROBE Statement—Checklist of Items for Cross-Sectional Studies

Comment 8

“Body mass index (BMI) was calculated as body weight (kg) divided by the square of height (m).”: Although the reader knows how BMI is calculated, a citation should be included.

Response 8
We appreciate the reviewer’s attention to detail. Although BMI calculation is widely recognized, we agree that providing a citation enhances academic rigor and acknowledges the authoritative source of this measurement standard. We have therefore added an appropriate reference (World Health Organization, 2000. “Obesity: Preventing and Managing the Global Epidemic. Report of a WHO Consultation, WHO Technical Report Series 894”) (line 113).

Added Reference

  1. Obesity: preventing and managing the global epidemic. Report of a WHO consultation. World Health Organ Tech Rep Ser 2000, 894, i-xii, 1-253.

Comment 9

“2.4. Biochemical Markers”: Were all of these parameters determined on fasting? If so, please specify.

Response 9

Thank you for raising this critical methodological point. We have revised the Methods to clarify that all biochemical parameters were measured from venous blood drawn after an overnight fast of at least 8 hours in accordance with KNHANES procedures. Relevant references have been added.

Line 134-137
“Biochemical measurements were obtained from venous blood after an overnight fast of ≥8 hours as per KNHANES procedures [21,22]. The samples were promptly processed, refrigerated, and delivered to the central laboratory of the Seegene Medical Foundation in Seoul.”

Results

Comment 10

Lines 166-187: You should split the paragraph into two. Paragraphs this long are difficult to read.

Response 10

We appreciate this helpful suggestion. We have revised the Results section by dividing the long paragraph into two shorter paragraphs to improve readability and flow.

Comment 11

Table 1. Table titles usually go above the table, not below.

Response 11

Thank you catching this formatting issue. We have corrected the formatting so that the title of Table 1 is placed above the table, in line with journal style guidelines.

Comment 12

“This suggests that despite an overall decline in carbohydrate consumption, older adults still depend more heavily on carbohydrates compared to younger generations.”, “These trends highlight generational differences in the consumptions of macronutrients, suggesting that young adults are shifting toward high-protein, low-carbohydrate diets more rapidly than the older adults.”, etc etc. Results should describe results, not interpretations or hypotheses; that's what the discussion is for. This is important for two reasons. First, as I said, results should focus on the obtained data. Second, eliminating information that belongs in other sections makes this section easier to read. Review the results section to ensure that all text belongs there and not in the discussion section.

Response 12

We agree with the reviewer’s observation. In the revised manuscript, we have carefully reviewed the Results section and removed interpretative sentences. These statements have been transferred to the Discussion section, ensuring that the Results now present only the obtained data, while interpretation is reserved for the Discussion.

Comment 13

Likewise, the authors begin with “Figure 1 illustrates the temporal trends in vital signs,”. However, it would be appreciated if, throughout the text, you could include “Figure 1-A”, “Figure 1-B”, etc., to help the reader find their bearings more easily.

Response 13

We appreciate this practical suggestion. We have revised the text to explicitly reference sub-panels (Figure 1-A, 1-B, 1-C, 1-D) throughout the Results section to facilitate easier cross-referencing between the text and figures.

Line 210-212
“Figure 1 illustrates the temporal trends in vital signs & anthropometric measures (Fig 1-A), nutrient intake (Fig 1-B), biochemical markers (Fig 1-C), and lifestyle behaviors (Fig 1-D) between 2013 and 2022 across young, middle-aged, and older adults.”

Discussion

Comment 14

The authors did an excellent job in the discussion; it's clear they've tried to explain all the results obtained. I only have two questions.:

Response 14

We sincerely thank the reviewer for the positive evaluation of our discussion. Please see below for our responses to your specific questions.

Comment 15

“4.1. Summary of Key Findings”: This section is too long. It should summaries the main findings of your study in no more than 15 lines.

Response 15

We agree with the reviewer’s suggestion. In the revised manuscript, we have shortened the “4.1. Summary of Key Findings” section so that it concisely highlights the principal findings within 15 lines. Redundant details were removed, and interpretive points have been streamlined, ensuring that this section serves as a clear summary.

Line 293-309

4.1. Summary of Key Finding

“In this nationally representative study of Korean adults using KNHANES 2013–2022 data, we found that total energy and carbohydrate intake declined steadily across all age groups, while protein, fat, saturated fat, and cholesterol intake increased significantly. Although absolute intake of lipid-related nutrients was highest among younger adults, the relative rate of increase was greatest in older adults. Carbohydrate intake remained highest in older adults and lowest in younger adults, whereas protein intake showed the opposite pattern.

These dietary changes were accompanied by modest increases in BMI, waist circumference, and SBP, with younger adults showing steeper rises than older adults. Biochemical markers displayed age-specific divergences: total cholesterol increased most in younger adults but decreased in older adults, while triglycerides declined in middle-aged and older adults.

Behavioral indicators also shifted markedly. Breakfast skipping rose sharply, particularly among young adults, exceeding 40% by 2022. Eating out, originally most frequent among young adults, showed a pronounced decline, with the steepest reductions during the COVID-19 pandemic. Sedentary time increased across all groups, with the greatest rise in older adults.

Comment 16

It would be interesting to include a paragraph (no more than 12–15 lines) explicitly comparing your results with those of other world regions (America, Europe, etc.). In other words, it would be interesting to see whether the same results are obtained in other countries.

Response 16

Thank you for your helpful suggestion. We have added new content comparing our findings with those from US and European studies. The revised text mentions that similar trends were observed globally: US adults reduced their intake of total energy and low-quality carbohydrates while increasing their consumption of plant-based protein and polyunsaturated fats. Meanwhile, European cohort studies also reported decreased carbohydrate intake alongside increased protein and fat consumption. This additional content provides an international context consistent with the reviewer's recommendation.

Line 316-328

4.2. Comparison with Previous Studies

“From 1999 to 2016, dietary energy intake among U.S. adults also showed a broadly similar trend. The proportion of low-quality carbohydrates decreased significantly, while the proportion of high-quality carbohydrates, plant-based protein, and polyunsaturated fats increased significantly. Despite improvements in key nutrient composition and diet quality, persistently high intakes of low-quality carbohydrates and saturated fat remained[25]. Moreover, longitudinal cohort studies in Europe have also confirmed shifts toward reduced carbohydrate intake and increased fat and protein consumption. For example, Winkvist et al. (a 10-year Swedish cohort) reported that dietary changes over a decade were characterized by decreased carbohydrate intake and increased protein and fat intake [16]. Similarly, the Amsterdam Growth & Health Longitudinal Study documented significant temporal effects on energy intake and the composition of major nutrients (protein, fat, carbohydrates) during a 20-year follow-up period [26].”

Reviewer 2 Report

Comments and Suggestions for Authors

This is a well-organized research study with adequate novelty wjich examine the nutrient intake, biochemical indicators, meal-related behaviors, and sedentary time among Korean adults, stratified by age group in the last decade. However, some points should be addressed.

  • An introductory statement of 1-2 sentences could be useful for the readers before reporting the main objective of the study in the Abstract.
  • The Introduction section is too short. The authors should report findings from other countries, mainly from Asia (e.g. Japan, China), as well as from high income countries with aged population (e.g. European countries).
  • The authors should specify whether weight and height for the calculation of BMI were measured or well sel-reported. If the data were self-reported then the authors should report recall-bias as a limitation of the study.
  • The data of the study were produced between 2013-2022. However, during the last years of this period, Covid-pandemic and lockdown was taken place. Are the data of the period 2020-2022 were examined separately since Covid pandemic may act as a counfounding factor? Are there any differences concerning this period and the previous period between 2013-2019? The authors should add some relevant discussion in the Discussion section.
  • Nutrient intake data were collected through face-to-face interviews conducted by trained professionals, including registered dietitians. This is a strong strength of the study that should be reported in the relevant section of the discussion section.
  • For sedentary behavior, the IPAQ questionnaire could be used as a validated tool and not merely simple question. This is a limitation of the study.

Author Response

We sincerely appreciate the time and effort you have dedicated to reviewing our work and for providing valuable insights to enhance the quality and impact of our study. Responses to the reviewers' comments and changes in the revised manuscript have been highlighted in red.

Reviewer 2 Comments

This is a well-organized research study with adequate novelty which examine the nutrient intake, biochemical indicators, meal-related behaviors, and sedentary time among Korean adults, stratified by age group in the last decade. However, some points should be addressed.

Response

We appreciate the time and effort you have dedicated to reviewing our work and for providing valuable insights to enhance the quality and impact of our study. Responses to the reviewers' comments and changes in the revised manuscript have been highlighted in red.

Comment 1

An introductory statement of 1-2 sentences could be useful for the readers before reporting the main objective of the study in the Abstract.

Response 1

We appreciate this suggestion. In the revised Abstract, we added a concise background sentence highlighting Korea’s rapid demographic shifts and Westernization of dietary habits, to better contextualize the study objectives.

Line 10-11

“Background: Rapid aging and dietary Westernization in Korea have raised concerns about shifting nutrition and lifestyle patterns.”

Comment 2

The Introduction section is too short. The authors should report findings from other countries, mainly from Asia (e.g. Japan, China), as well as from high income countries with aged population (e.g. European countries).

Response 2

Thank you for this valuable suggestion. We have expanded the Introduction section to include evidence from other regions, particularly Japan, China, and several European countries experiencing rapid population aging. The revised text emphasizes that similar nutritional and lifestyle changes—characterized by reduced carbohydrate intake and increased fat and protein consumption—have also been observed in these countries. This additional content provides a broader international context and strengthens the rationale for examining these trends within the Korean population.

Line 60-76

“Previous studies from various countries have provided important insights into recent shifts in dietary and lifestyle patterns. In Japan, analyses from the 2003–2019 National Health and Nutrition Survey demonstrated a clear dietary pattern transition, characterized by a gradual decline in carbohydrate intake and increases in fat and protein consumption, alongside improvements in overall diet quality [14]. Similarly, data from the China Health and Nutrition Survey revealed substantial changes in macronutrient composition between 1991 and 2011, with notable increases in total fat and animal-source foods and a decline in carbohydrate intake, largely driven by urbanization and socioeconomic development [15]. European cohort studies have also documented trends toward higher fat and protein intake and lower carbohydrate in-take over time [16].

Despite these global findings, there remains limited evidence on how such nutritional and behavioral transitions have occurred across different generations in Korea. Furthermore, the absence of integrated, age-stratified analyses limits our understand-ing of how these changes differ across generations and whether recent events such as the COVID-19 pandemic may have accelerated or altered these trends. Reducing this gap is crucial for developing personalized public health strategies and age-specific interventions to promote healthy aging and reduce the risk of chronic diseases [17,18].”

Comment 3

The authors should specify whether weight and height for the calculation of BMI were measured or well self-reported. If the data were self-reported then the authors should report recall-bias as a limitation of the study.

Response 3

We thank the reviewer for this useful comment. In the revised manuscript, we have clarified that both height and weight were directly measured by trained personnel according to standardized KNHANES examination protocols, rather than self-reported. The corresponding sentence in the Methods section has been revised accordingly.

Line 112-115
“Body mass index (BMI) was calculated as body weight (kg) divided by the square of height (m) [19], with both height and weight measured directly by trained examiners according to standardized KNHANES protocols.”

Comment 4

The data of the study were produced between 2013-2022. However, during the last years of this period, Covid-pandemic and lockdown was taken place. Are the data of the period 2020-2022 were examined separately since Covid pandemic may act as a counfounding factor? Are there any differences concerning this period and the previous period between 2013-2019? The authors should add some relevant discussion in the Discussion section.

Response 4

We thank the reviewer for this insightful comment. The potential impact of the COVID-19 pandemic was carefully considered in our analysis. As shown in Figure 1, the study period was divided into four distinct intervals (2013–2015, 2016–2018, 2019–2021, and 2022) to visualize temporal trends, allowing the period overlapping with the pandemic (2020–2022) to be distinguished from the pre-pandemic years (2013–2019). Moreover, Figure 2 specifically compares changes between 2013–2015 and 2020–2022 to capture the net differences before and during the pandemic.

Our results indicate that while macronutrient trends followed gradual long-term trajectories, certain behavioral indicators—particularly eating-out frequency and sedentary time—showed more abrupt changes during 2020–2022, likely reflecting social-distancing restrictions and lifestyle disruptions. To clarify this interpretation, we have expanded the Discussion section to highlight that the COVID-19 period was analyzed separately and may have contributed to short-term fluctuations in these behaviors.

Line 410-415

“The period 2020–2022, corresponding to the COVID-19 pandemic, was analyzed separately to account for its potential confounding effects. Behavioral indicators such as eating-out frequency and sedentary time showed more pronounced changes during this period, reflecting pandemic-related restrictions, whereas macronutrient composition trends appeared to follow longer-term secular shifts.”

Comment 5

Nutrient intake data were collected through face-to-face interviews conducted by trained professionals, including registered dietitians. This is a strong strength of the study that should be reported in the relevant section of the discussion section.

Response 5

We appreciate the reviewer’s positive comment and agree that this methodological feature deserves explicit recognition We fully agree that the use of face-to-face interviews conducted by trained professionals, including registered dietitians, represents a major methodological strength of our study that distinguishes it from studies relying on self-administered questionnaires or online surveys. To address this, we have added a sentence in the Strengths and Limitations section of the Discussion explicitly emphasizing this point and its implications for data quality.

Line 434-438
“A key methodological strength of this study is that dietary intake data were collected through face-to-face interviews conducted by trained professionals, including registered dietitians, which ensures high data quality and measurement reliability and minimizes common sources of error such as misinterpretation of questions or inaccurate portion size estimation.”

Comment 6

For sedentary behavior, the IPAQ questionnaire could be used as a validated tool and not merely simple question. This is a limitation of the study.

Response 6

We thank the reviewer for this helpful comment that identifies an important limitation of our study. In the KNHANES survey, sedentary behavior is assessed using a standardized single-item question that asks participants to report their average daily sitting time, covering multiple contexts such as work, home, academic, and leisure activities. Although the International Physical Activity Questionnaire (IPAQ) is a validated instrument, it was not included in the KNHANES protocol during the study period. To address this point, we have added a statement in the Limitations section of the Discussion acknowledging that the use of a single self-reported item, rather than a multi-item validated questionnaire such as IPAQ, may have led to measurement bias and may have limited our ability to capture the full complexity of sedentary behavior patterns.
Line 447-453

“In addition, sedentary behavior was assessed using a single self-reported question rather than a validated multi-item instrument such as the International Physical Activity Questionnaire (IPAQ), which may have introduced some degree of measurement bias and may have resulted in underestimation or overestimation of actual sedentary time. Future studies would benefit from incorporating comprehensive validated instruments to capture more nuanced patterns of sedentary behavior across different contexts and time periods.

Round 2

Reviewer 1 Report

Comments and Suggestions for Authors

Comments to authors

The authors correctly addressed all the issues raised.

Reviewer 2 Report

Comments and Suggestions for Authors

The authors have considerably improved their manuscript.